# Enablement of nursing home residents in infection prevention during general practitioner visits: A qualitative study

**Judith Hammerschmidt[1], Lina Heier[1,2], Nicole Ernstmann**[1,2]*

**1** Institute for Patient Safety, University Hospital Bonn, Bonn, Germany, **2** Department for Psychosomatic Medicine and Psychotherapy, Center for Health Communication and Health Services Research, University Hospital Bonn, Bonn, Germany

* nicole.ernstmann@ukbonn.de

## Abstract

### Introduction

Hand hygiene measures are essential to protect nursing home residents against nosocomial infections. Evidence on the prevention of nosocomial infections for nursing home residents by general practitioners during their medical visits in nursing homes or how they enable nursing home residents to perform hand hygiene measures is lacking. This study aimed to explore hand hygiene behaviors of general practitioners in nursing homes, their attitudes toward infection prevention measures, and the enablement of nursing home residents in performing hand hygiene measures.

### Materials and methods

Semi-structured interviews were conducted with general practitioners and nursing home residents in Germany. Interviews were audio-recorded and transcribed. Data were analyzed using thematic content analysis.

### Results

Overall, 12 general practitioners and 12 nursing home residents participated in the study. The general practitioners expressed the fact that the possibilities for practicing hand hygiene differ in individual nursing homes. For nursing home residents, the availability of hand rub solutions was limited. Instructions for residents on hand disinfection from general practitioners was not described. Due to the lack of enablement, the residents' knowledge on how to correctly perform hand hygiene was low, although some of the nursing home residents have experience with multidrug-resistant organisms. The nursing home residents varied in their needs for active participation and enablement during the general practitioners visit.

### Conclusion

Nursing home residents require continuous enablement by their general practitioners to maintain adequate hand hygiene. Therefore, general practitioners should consider the

**Data Availability Statement:** All datasets of study participants are stored in the archive of the University Hospital Bonn, Germany, according to data protection requirements. There are ethical

restrictions on sharing the present study data publicly. Data cannot be shared publicly because of privacy reasons. Data requests can be forwarded to the attention of the Ethics Committee of the University-Hospital-Bonn: ethik@ukbonn.de.

**Funding:** The present study was supported by Bundesministerium für Gesundheit in the form of a grant (No. IIA5-2512FSB105); the funding from this source was received by JH. The funders had no role in study design, data collection and analysis, decision to publish, or preparation of the manuscript.

**Competing interests:** The authors have declared that no competing interests exist.

**Abbreviations:** GP(s), General practitioner(s); HAI, Healthcare-acquired infection; I, Interviewer (within the interview passages; R, Nursing home resident (within the interview passages); WHO, World Health Organization.

different needs of nursing home residents to ensure adequate individual hand hygiene and safety for the residents. Existing guidelines for infection prevention and control do not adequately cover the nursing home care setting for the enablement of residents to enquire about hand hygiene.

## Introduction

Healthcare-associated infection (HAI) is one of the most severe global public health problems, with 16 million deaths per year [1,2]. The European Center for Disease Prevention and Control estimates that approximately 4.4 million patients acquire an HAI each year in the 27 European member states, and approximately 37,000 deaths result directly from these infections [3,4]. *Staphylococcus aureus* infections are one of the three most common antimicrobial-resistant pathogens [1]. Methicillin-resistant *Staphylococcus aureus* (MRSA) is a type of staphylococcus bacteria that is resistant to many antibiotics [1].

In a healthcare setting, such as nursing homes, MRSA can cause serious infections, such as bloodstream infections and pneumonia, which can lead to sepsis and death [3]. MRSA spreads via the hands of healthcare providers that have been contaminated after touching an infected wound or a contaminated surface. Furthermore, asymptomatic individuals with MRSA can spread the bacteria to others [5]. In 2009, the "Council Recommendation on patient safety, including the prevention and control of HAI" invited the member states to adopt specific strategies on the prudent use of antimicrobial agents with the aim of improving patient safety [6]

In 2017, 3.4 million people were care-dependent as per the definition of the German "Care Insurance Act" [7]. The people are often frail due to age-related chronic diseases. They have complex risk profiles for infections and antibiotic treatment and require special protective isolation measures [8,9]. The infections within these vulnerable populations often lead to suffering, frailty, or death [10,11].

The most effective single measure for infection prevention in various healthcare settings is antiseptic hand rub [12–14]. Antiseptic hand rub inhibits the growth of microorganisms, and compared with hand washing, no other resources, such as water and towels, are needed [15]. In healthcare facilities, the use of alcohol-based hand sanitizers is mandatory because hand washing is not as effective and increases the risk of microbial transmission [16–19].

Studies show a lack of research on infection prevention measures and transmission paths in nursing homes [20–22]. The existing guidelines for infection prevention and control do not adequately cover the nursing home setting, and more research is needed to determine which interventions, such as patient/caregiver education, would be useful to prevent infections in this complex setting [10,20,23,24].

In Germany, most nursing home residents receive medical care almost exclusively from their GP. However, it is not mandatory for a GP to have an overview of all HAIs and all antimicrobial therapies for all nursing home residents [25]. This situation makes it difficult to establish consistent infection prevention and control measures in the work processes in nursing homes,[8,23] which the GP could follow [26]. The sharing of written healthcare information on aspects of infection prevention, control, and antibiotic prescription between healthcare professionals is not mandatory. GPs are not required to take any specialized training in geriatrics or infection prevention and control [27].

Enablement is a process by which the healthcare provider assists patients in recognizing, promoting, and enhancing their health [28]. Enablement in performing hand hygiene

measures could reduce the possibility of the chain reactions of cross-infections and spread to the environment. Little is known about whether and how nursing home residents are enabled regarding infection prevention measures, such as hand hygiene, by their GPs and nursing staff [29–31]. The World Health Organization (WHO) recommends the training of patients and their families on the use and indications of hand hygiene measures to reduce multidrug-resistant organisms [1]. However, thus far, nursing home residents have rarely been involved and enabled in hand hygiene measures on a routine basis [24]; they are instructed in the event of an existing chain of infection, such as a norovirus infection, but not preventively [32]. There is a lack of research on nursing home residents' enablement regarding hand hygiene.

This study aims to explore hand hygiene behavior of GPs in nursing homes, their attitudes toward infection prevention measures, and the enablement of nursing home residents in hand hygiene.

## Materials and methods

### Study design

This study was part of a more extensive interventional PränosInAA study (2012–2015) with a focus on improving hygiene practices and the rational use of antibiotics in nursing homes. Our study focused on cross-sectional, semi-structured, problem-based interviews with GPs and nursing home residents from the PränosInAA study [22].

### Recruitment and informed consent

A pool of 542 nursing homes were identified in the Rhineland Area, Germany. After a purposeful sampling process (i.e., nursing homes for the care of elderly residents with a mix of different care levels ranging from basic to full care, to meet all aspects of resident care, and a minimum of 80 residents per nursing home), six facilities were randomly selected and invited to participate in the project, all of which agreed to participate. These six participating nursing homes were located within a radius of 50 km of Bonn because they were visited weekly by two medical doctors of the PränosInAA study using an antibiotic stewardship program. All long-term nursing home residents were invited to participate in the PränosInAA project. At the beginning of the study, six information events were conducted by the researchers for 588 nursing home residents who were potentially interested in voluntary participation. The residents received oral and written information about data protection, voluntary participation, the aim of the study, the methods and the duration of in the study. Overall, 332 (56.5%) provided written informed consent for participation in the PränosInAA project. If the residents had no cognitive limitations, they gave their own written informed consent to participate in the PränosInAA study. If with cognitive limitations, a court-appointed guardian provided written informed consent. The participants in all six nursing homes gave their additional oral, recorded, and written informed consent directly before the interviews. The inclusion criteria for the purposeful interview sample of nursing home residents were age >65 years, permanent residence in a participating nursing home, and receipt of medical visits by a GP in the nursing homes. The exclusion criteria were diagnoses of cognitive impairment or speech or hearing disorders. Three researchers arrange appointments with nursing home residents. For the GP interviews, 250 GPs in the region of Bonn were invited to participate in the larger project. The inclusion criteria were accreditation with the statutory health insurance and regular medical visits in nursing homes. Twelve GPs provided additional informed consent for the interviews.

## Data collection

The 12 interviews with nursing home residents were conducted in 2015 from three female researchers; JH and SE have both master degrees in nursing management and are registered nurses with 10 years of practical work experience in nursing care. NC has Master a degree in health care management and one year of practical work experience in nursing care.

Geriatric nurses introduced the interviewers to the residents because they previously had no direct contact or relationship. The geriatric nurses were known by the residents and explained again the aim of the study, voluntary participation, duration, content of the conversations, and data protection measures in the absence of the interviewers. This approach ensured that the residents did not feel restricted in their decision-making as they were in a state of dependence on their GP. The interview guide, based on our initial literature review. Interview questions from the interview guide were kept as open as possible to allow the residents to answer according to their need for self-protection and maintain well-being. In each case, two researchers visited the residents in their rooms (JH, NC, SE). The interview process was based on a semi-structured interview design with 15 questions for the nursing home residents (see S1 Appendix). As the interview may be perceived as a stressful situation for the nursing home residents, the interview guide was not piloted due to ethical reasons. Nevertheless, the appropriateness of the interview guide was critically observed during the interviews and judged as adequate. The interviews were conducted on a one-to-one basis, audio-recorded, and transcribed verbatim in German; the second researcher observed the interview and wrote field notes about how the interviewees talked about different aspects. After each interview, the researchers validated the incremental information gathered. After 12 interviews, the researchers established they had sufficient data saturation as only a little incremental information was gathered through additional interviews. Data were sufficient to allow category formation. The category formation was formed in a continuing concentration process during which five main categories were fixed. They were elaborated on by linking the research questions to the focused content. In doing so, relating these five categories to the categories established by the GP interviews was also possible. To avoid changes in meaning and interpretation, the original codes were translated from German to English by a professional translation service in the final step of processing the results.

To start the conversation, the residents were asked to talk about the medical care they needed before they were admitted to nursing home. This procedure was examined to simplify the comparison of the residents' life situation before admission and afterward. Later, they talked about their experiences with infection prevention measures during GP visits. The interviews lasted 9–16 minutes. Sociodemographic information was collected at the end of the interviews.

The 12 interviews with GPs were conducted in 2015. GPs were visited in their offices by an interviewer (JH, NC, SE). At the start, interviewees were written and oral informed about the voluntary participation, data protection, possibility of termination at any time, aim of the study, and duration and content of the interviews. The GP gave their oral, recorded, and written informed consent directly before the interviews. The GP interview guide (see S2 Appendix), based on our initial literature review. The interviews consisted of 11 general questions, in which GPs could report on their experiences with infection control measures during visits to the nursing homes, their daily experiences with hand hygiene during visiting rounds, and surgical dressing changes. The interview process was based on a semi-structured interview design. The interviews lasted 9–19 minutes. Sociodemographic information was collected at the end of the interviews.

## Data analysis

All interviews with nursing home residents and GPs were audio-recorded, anonymized, transcribed verbatim, and coded (MAXQDA version 11; Copyright ©1995–2017, VERBI GmbH).

Data analysis began during data collection and was an ongoing process. The derived thematic content was independently coded [33] by the same three researchers who conducted the interviews. The research team used an open and selective coding process to identify and characterize text units from each conversation [34]. The field notes helped identify meaningful, expressive phrases, pauses, body language, and emotions in interview passages during the coding process. Five major categories were developed in line with the research questions from the PränosInAA study:

1. Perceived organizational commitment to enable hand hygiene practices by developing factors related to structures and processes hindering or facilitating them (e.g., access to alcohol-based hand rubs and medical gloves).

2. Perceived organizational management of hygiene issues (e.g., communication between nursing home residents, nursing staff, and GPs during medical visits).

3. Self-reflection of GPs regarding their compliance with hygiene standards.

4. Perceptions of GPs and nursing home residents regarding their knowledge of hand hygiene.

5. Perceptions of GPs and nursing home residents concerning enablement of hand hygiene.

Relevant categories were discussed by a multidisciplinary team of researchers from the fields of health management, nursing science, and psychology. Any discrepancies of themes were discussed and resolved by consensus. The transcripts or results of data analysis were not discussed with the study participants themselves.

## Results

The nursing home residents' average age was 82 years; four were male, and eight were female. The residents suffered from a variety of chronic diseases, such as diabetes mellitus, osteoarthritis and osteoporosis, cardiovascular diseases, and chronic obstructive lung disease. The average age of the GPs was 48 years; 4 out of 12 were female. The GPs had all worked as specialists for >5 years in their medical practices, and they visited patients in nursing homes regularly. In summary, the content analysis enabled the construction of five main categories to explore the perceived hand hygiene behavior of GPs and nursing home residents, their attitudes toward infection prevention measures, and the perceived enablement of nursing home residents in hand hygiene behavior during visits. Subsequently, results are reported for each major category with examples from interview transcriptions.

### Perceived organizational commitment to enable hand hygiene practices by developing factors related to structures and processes hindering or facilitating them (e.g., access to alcohol-based hand rubs and medical gloves)

The GPs explained that the more nursing home residents they visit, the more scheduled and structured their visits and prescriptions in the nursing homes are, for example, with a fixed schedule for visits on one or two afternoons a week. In cases of specific questions, GPs prefer to have an accompanying nurse and access to alcohol-based hand rub and medical gloves. The same applies to the deterioration of health or after a hospital stay, especially for residents with

dementia. They expressed that nurses do not often have the time to accompany the GPs during a visit. Some GPs argued that regular visits make it easier for them to coordinate with the nurses and the residents. In cases when a nurse could accompany the visiting GP, the diagnosis and indications for medical prescriptions could be documented directly at the same time in both the residents' charts at the nursing home and the GP's medical file.

> *I: Is there usually a nurse at the nursing home to assist you during your visits?*
>
> *GP (male, 22 years of experience): I will make sure of that! Better: I insist on it, >laughs <let us put it like this.*
>
> *I: In some cases, the GPs could not find a nurse to accompany the visits. Do you know that too?*
>
> *GP: That is very problematic. They are all in the resident rooms, and you lose much time trying to find someone. The residents are all in relatively poor health. I cannot get anywhere without the help of a nurse because the majority of patients have dementia.*

Some GPs reported visiting nursing home residents only for acute emergency calls due to better billing options. In this case, they accept not being actively accompanied by nurses. From the GP's perspective, there are no regulations as to whether and how the visit is documented in the resident's and the GP's medical files when no nurse accompanies the visit. In cases of infections, medical prescriptions and therapy were often documented only in the GP's medical file, sometimes without informing the nursing staff or documenting it in the resident's file.

## Perceived organizational management of hygiene issues (e.g., communication between nursing home residents, nursing staff, and gps during medical visits)

Regarding their behavior toward nursing home residents diagnosed with multidrug-resistant organisms, the GPs had to weigh the protection of the health of all nursing home residents against the freedom of the individual patient. All GPs saw themselves as mere advisors for maintaining the quality of life of the older people. At the same time, some were aware of the decision-making structures and responsibilities of nursing home management.

> *GP (male, 12 years of experience): If it is Clostridia or MRSA, then there is a standard in every nursing home for which the hygiene manager is responsible. That means I have nothing to do with their decisions. [. . .] Of course, I give advice. I always try to find the balance between freedom and isolation, which is not easy because people live there permanently.*

## Self-reflection of GPs regarding their compliance with hygiene standards

The GPs are external visitors into the nursing homes. For the residents, it can be very dangerous when visitors carry external pathological germs on their hands into the nursing homes. Therefore, GPs are advised in hygiene standards to disinfect their hands before, during, and after every single patient and medical visits. In the interviews, the GPs were asked about their hand disinfection options before and during visits to the nursing homes. They mentioned that the opportunities varied from institution to institution. Only some nursing homes install disinfectant dispensers in the corridors for all visitors. In some cases, they stated that disinfectant is available only upon request or in the wardroom. Therefore, some GPs use their portable hand disinfectant. Some GPs reported disinfecting their hands only in the wardrooms or

workrooms. All GPs reported that there were hardly any hand disinfection options in the residents' rooms and bathrooms. This could be important because there is direct contact from nurses and doctors with the residents in their rooms and bathrooms.

> *I: And do you carry hand rub with you when going for a visit? Or is disinfectant available in the nursing home?*
>
> *GP (male, 18 years of experience): No, it is right there, and I use it, yes. In nursing home X, there has now been a hand disinfectant at the front door for a year or so. That is for visitors to use.*
>
> *I: And do you use it?*
>
> *GP: Me? Usually not, no.*

Microbiological differentiation is used to determine the bacterium and the appropriate antibiotic therapy. If the bacterium is analyzed, it is important that the staff, residents, and relatives are aware of the therapy and the possible multi-resistances of the pathogen. They could initiate effective protective measures in the nursing home, such as hand hygiene, masks, gloves, and protective clothing.

In one case, a GP explained that she rejects microbiological differentiation of the pathogens. If they did not prescribe an effective and appropriate antibiotic therapy, this behavior can have very serious health consequences for the residents. The GP explained the behavior by referring to the quality of life for all the nursing home residents.

> *GP (female, 27 years of experience): For example, MRSA or ESBL. What beautiful things they are! I cannot lie now, can I? So, with MRSA, I do not think you have to isolate in a nursing home. Absolutely not! Otherwise, people have no quality of life at all. I am already in favor of disinfection, but there is no need to shut down the entire program. Moreover, I am not the type to take three swabs. We know it is serious, but it is also much dramatized.*

## Element of perceived behavioral control: perceptions of GPs and nursing home residents regarding their knowledge of hand hygiene

The participating nursing home residents noticed when the GP performed hand disinfection. They express their beliefs that their doctors behave correctly hygienically.

> *I: Can you remember if your GP disinfects his hands while he visits you?*
>
> *R: Yes, I can. When he comes in, he rubs his hands.*
>
> *I: And does he do this when he examines you?*
>
> *R: I think he will do it when he leaves.*

Some nursing home residents tried to be polite when asked whether GPs wore a protective gown and gloves during the rounds. The same applied to the question of whether the doctor disinfects his or her hands during the rounds in the room.

> *I: (. . . .) do you remember if your* GP *wear a doctor's coat when he comes here? Alternatively, does he come in regular clothes?*
>
> *R: He wears his regular clothes. No, white coat.*
>
> *I: And does he disinfect his hands when he is in here?*

*R: Oh, so! I certainly hope so!*

*I: Can you remember that?*

*R: I will watch him next time (laughs).*

*I: Do you dare to tell him?*

*R: No, he won't come back, or he won't like me anymore (laughs).*

*I: Maybe he'll be happy about your active participation and help.*

*R: Maybe. I don't think so.*

## Perceptions of GPs and nursing home residents concerning enablement of hand hygiene

The participating nursing home residents had various needs for active participation and enablement during their GP visits. Some took the opportunity to ask about "*everything*" that may be important to them in relation to medication and treatment. They felt "*in good hands*" during their visits when the GPs visited them more frequently for acute illnesses. The quality of medical care was also noticeable to nursing home residents when the GP came to visit regularly. It was clear to all interviewees that the GP leads the process and the consultation during visits.

*I: And can you ask your GP questions about your illness? About the effects and side effects of your medication?*

*R: About everything! He comes to the house. He's got several residents whom he needs to see, and afterwards, he comes to me. A few weeks ago, when I had a bad cold and was desperate due to a severe cough, he even came at nine o'clock in the evening, and there I had many questions.*

*I: Can you ask him questions about your illness?*

*R: He is an excellent doctor. He often comes to me.*

Nursing home residents were asked to report their experiences with hand hygiene during the visits. None of the residents described receiving hand hygiene training. None of the interviewees expects their GP to show them preventive behavior. The residents' expectations focused on the therapy of diseases and regular visits. They ask if the GP has time for preventive consultations and if it is the GP's role.

*I: Did your doctors or nurses in the hospital show you how to disinfect your hands?*

*R: No. (. . .).*

*I: Has a GP ever shown you how to disinfect your hands?*

*R: (laughs) No, do they do that? They don't have time! (laughs).*

Although all nursing home residents reported that they received no hand disinfection instructions from their GPs or during hospital stays, they were able to explain why this measure is essential for infection prevention. During the conversation, they reflected on this sensible measure for themselves and other residents. They became aware that with hand hygiene, they protect not only themselves but also other residents.

*I: Did the GP talk to you about hand hygiene, or did he show you how you could do it?*

*R. No, not that I know. Would it be okay if he showed it to me?*

*I: (. . .) that way, you could better protect yourself from infections.*

*R: Yes, that would be good for me, but also for the others here who are even sicker than I am.*

Residents did not feel able to talk to their GP about a lack of hand disinfection. None of the residents described receiving hand hygiene training or an explanation of why this is important to their health. However, the interviewees understood that this approach could protect them from infections. They reflected that if they were trained in hand hygiene before and during hospital stays or in nursing homes, their behavior could protect them and others from HAIs. If the GPs do not disinfect their hands, the residents explained it politely as due to a lack of time, never with carelessness, lack of knowledge, lack of opportunity, or bad habits. The same was true for not wearing a gown for visits. They were afraid that the GP would no longer like and visit them if they brought up this issue.

## Discussion

Due to the increasing global problem of multidrug-resistant organisms and the global COVID-19 pandemic, the awareness of preventive hygiene measures, such as hand washing and alcohol-based hand rub, remains an important topic. This qualitative study was conducted before the COVID-19 pandemic. The study aimed to explore the hand hygiene behavior of GPs in nursing homes, their attitudes toward infection prevention measures, and the enablement of nursing home residents in performing hand hygiene measures.

This study found a lack of nursing support during GP visits and highlighted the consequences. To receive nursing support during the visits, an appointment between the GP and the nursing staff is required. This regular support can be beneficial for both professions and the residents themselves, as they receive regular care [22].

Due to the legal freedom of medical therapy, GPs in Germany are free to decide on the frequency, duration, scheduling, and execution of their medical visits to nursing homes [25]. The nursing home residents perceive the regular visits as a quality criterion for their GPs. In this study, the residents did not assess the content of the visits in terms of training and health literacy. Maintaining a positive relationship with their GP was more important to them. The GP is supposed to come when they need medical help—this relationship of trust and dependence is evident in all the interviews. The residents trust the GPs to do their job well. In the case of infections, it is important to monitor this process from GPs and nursing staff as early as possible [35]. This overall situation makes it difficult for the nursing home managers to establish consistent infection prevention and control measures in the work processes during visits to nursing homes [8,24].

Sharing written healthcare information on aspects of infection prevention, control, and antibiotic prescription between healthcare professionals is not mandatory. The nursing staff have daily contact with the residents and can assess the altered state of health. Nurses can describe the patient's symptoms, and GPs can base early diagnosis and treatment on this [35]. Many residents have cognitive impairments and are especially dependent on the care and attention of health professionals. This attention helps them clarify current health problems, such as signs of infection, with their GP. In this study, the GPs described different time spans and organizational forms of nursing home visits, which is consistent with the results of previous studies in Germany [27,36]. The GPs' arguments for their preferred type of visit ranged from financial considerations when settling the service with the health insurance companies,

e.g., for spontaneous visits in the event of acute deterioration of the patient's condition, to strict visit schedules and a preference for routine support by the nursing staff. GPs visit shortly after hospitalization to assess the patient's condition and adapt the medication plan to the health status of the patient. When GPs treated more nursing home residents, it was relevant for them to exchange information about patients and suggest treatments with nursing staff [36]. GPs are not required to take any specialized training in geriatrics or infection prevention and control [27]. The fact that German health insurance pays coordinated procedures less than ad hoc visits is highly questionable from the infection prevention perspective [10,18]. Ad hoc visits must be for an acute illness or a change in the state of health. From the perspective of nursing home residents who do not have cognitive impairment and want to make an appointment with their GP, fixed appointments help them prepare for visits and be in their rooms [27,36].

Nursing home residents reported that GP visits gave them "fatherly" and calm emotional support when GPs regularly asked about their health condition. In contrast to Sak et al. (2017) and Fleischman et al. (2016), in the interviews, the GPs' consulting function was not reported, nor was a mutual conversation on eye-level described by both sides [27,37]. Because viruses and bacteria are not visible to the eye, preventive hygienic measures must be taken.[38]. In the interviews, GPs interpreted the hygiene guidelines more freely. For example, the use of hand disinfection according to the WHO's "5 Moments of Hand Hygiene" might indicate a reflection on the GP's own role. In an interview, the GP explained that he does not disinfect his hands when he visits the nursing home because he does not see himself as a "visitor". The same applies to the decision to not record the three swabs in the neck–nose–throat area from residents with MRSA. Here, a deliberate argument was made against the guidelines. Such perceived behavior may not be conscious and should be investigated in further studies. Preventive measures such as hand hygiene could interrupt infective chain reactions [39]. The treatment is based on the principles of hygiene [12,40]. In this study, the GPs did not instruct and train residents hand hygiene. At the beginning of the COVID-19 pandemic, when there were only preventive measures and no vaccinations, paying attention to and understanding the importance of preventive hygiene rose worldwide. Therefore, public and governmental international health organizations (e.g., the WHO, Centers for Disease Control and Prevention, Robert Koch Institute) provided daily information and instructions. In nursing homes, the professionals and residents without mental disabilities made great efforts to follow these instructions, e.g., quarantines and visiting bans [41]. This was particularly difficult at the beginning of the pandemic, as medical supplies and disinfectants were not available in sufficient quantities. In Germany, the number of HAI increased in 2020 [39]. HAI continuously rose during the COVID-19 pandemic, but public awareness has reduced. This can be attributed to the insufficient availability of medical devices and hand disinfectants and the lack of nursing staff [39].

Valensi et al. (2008) focused on patients >70 years of age with type 2 diabetes and similarly found that "caring relationship" was more important than "active participation in decision-making" [42]. In the interviews in this study, the GPs did not report consistent compliance with regulations regarding hygiene during their visits. The GPs were aware of the regulations from the quality manuals and professional exchanges with other colleagues, and two GPs were involved in the preparation of the national guidelines for the treatment and recording of MRSA in nursing homes. However, the implementation of hand hygiene during visits was interpreted differently. None of the nursing home residents were introduced to, or made aware of, infection prevention behavior by their GPs, not even during or after multidrug-resistant infections, which are often the reason for very severe health restrictions, frailty, and deaths [1,2]. The nursing home residents did not describe any active involvement or receipt of instructions for hand disinfection from nursing staff or GPs. Sak et al. (2017) reported that

two-thirds of their sample of older patients were satisfied with their current involvement in medical decision-making and that this group may also have a moderate or lower level of health literacy [37]. The process of active involvement in decision-making processes is often unfamiliar to older patients [37]. The nursing home residents were somewhat uncertain about their expectations regarding active consultation with GPs; the expenditure of time seemed unrealistic to them. Their focus was on the GP's reliability in the case of acute illnesses. The task of the GPs to involve and enable the patient was perceived as appropriate by the nursing home residents. The residents were not only concerned about their health but also infection prevention among the other residents.

GPs described their hand hygiene behavior as being influenced by the availability of hand rub during visits and their perception of infection risks, especially when a resident had multi-drug-resistant infection. GPs did not mention complying with the 5 Moments for Hand Hygiene [43]. Improvements in nursing homes are often hindered by the prevailing conflict between maintaining a homelike environment and a higher standard of living on the one hand and adopting and monitoring state-mandated infection prevention measures on the other [44]. However, infection prevention in nursing homes is vital due to patient proximity and multi-morbidity, as well as multidrug-resistant pathogens. Nursing home residents reported either themselves or someone in their immediate environment having had multidrug-resistant infections with permanent health consequences or death.

There are limitations to be considered in interpreting the findings of this study. The results should be considered as indicatory since this exploratory study was conducted based on 24 interviews with purposefully selected interview partners. There may be a selection bias due to the interviewees' interest in the study. The results cannot be generalized to nursing home residents with cognitive impairments, impaired consciousness, or extensive nursing care needs. These nursing home residents are particularly dependent on medical staff to show them Apreventive measures, guide them, and provide protection through their own preventive measures. Being constantly aware of this responsibility is very challenging and requires a high level of professionalism. The perspectives, perceived behavior, knowledge, and attitudes of hand hygiene measures from nurses and nursing home managers in the PränosInAA study was published before this study [22]. Social desirability effects might have biased the answers of nursing home residents and GPs. However, the selection criteria, possible selection bias, or social desirability bias might have led to an underestimation of the hand hygiene deficits, not to an overestimation.

## Conclusions

This study revealed major gaps in hand hygiene compliance on both the GPs' and the nursing home residents' sides. These deficits emerged in perceived knowledge, attitudes, and perceived behaviors. The asymmetrical paternalistic relationship between nursing home residents and their GPs makes it difficult for nursing home residents to speak up for their concerns. Patient involvement in preventive hygiene measures must become more pronounced during the COVID-19 pandemic. The idea of the patient element has not yet received necessary attention, especially in nursing home care. Not every resident has a cognitive impairment that might prevent them from involvement.

Further research into the COVID-19 pandemic should be conducted on the enablement of older people. The role model function of healthcare professionals and family involvement should also be considered in the development of training programs. Continuous improvements in infection prevention in nursing homes can only succeed if internal and external participants, such as nursing home residents and GPs, adhere to established hand hygiene standards.

## Supporting information

**S1 Appendix.**
(TIF)

**S2 Appendix.**
(TIF)

## Acknowledgments

The authors would like to thank all participating nursing home residents and the GPs for their trust and support during this study. We also thank our research team from the PränosInAA project under the scientific lead of Martin Exner, Institute for Hygiene and Public Health, University Hospital Bonn, and our colleagues from the Institute for Patient Safety, University Hospital Bonn, for their motivating and constructive support. We thank Tanja Manser for her helpful contributions to prior versions of this manuscript. We would especially like to thank student assistants Nasanin Chenari, Sandra Eicker, Marie Debrouwere, and Donia Riouchi for their efforts and commitment during this project.

## Declarations

### Ethics statement

Ethics approval according to the guidelines of the Declaration of Helsinki was obtained from the Ethics Committee of the University Hospital Bonn in December 2012 (reference number 069/12).

### Availability of data and materials

All datasets of study participants are stored in the archive of the University Hospital Bonn, Germany, according to data protection requirements. There are ethical restrictions on sharing the present study data publicly. Data cannot be shared publicly because of privacy reasons. Data requests can be forwarded to the attention of the Ethics Committee of the University-Hospital-Bonn: ethik@ukbonn.de

## Author Contributions

**Data curation:** Judith Hammerschmidt.

**Formal analysis:** Judith Hammerschmidt.

**Investigation:** Judith Hammerschmidt.

**Methodology:** Judith Hammerschmidt, Lina Heier, Nicole Ernstmann.

**Project administration:** Judith Hammerschmidt.

**Supervision:** Nicole Ernstmann.

**Writing – original draft:** Judith Hammerschmidt.

**Writing – review & editing:** Lina Heier, Nicole Ernstmann.

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
