## [Decision Letter · Decision Letter 0]

2 Aug 2021

PONE-D-20-34748

Enablement of nursing home residents in infection prevention during GPs visits: A qualitative study

PLOS ONE

Dear Dr. Ernstmann,

Thank you for submitting your manuscript to PLOS ONE. After careful consideration, we feel that it has merit but does not fully meet PLOS ONE’s publication criteria as it currently stands. Therefore, we invite you to submit a revised version of the manuscript that addresses the points raised during the review process.

We look forward to receiving your revised manuscript.

Kind regards,

Monika Pogorzelska-Maziarz

Academic Editor

PLOS ONE

Journal Requirements:

2. When reporting the results of qualitative research, we suggest consulting the COREQ guidelines: http://intqhc.oxfordjournals.org/content/19/6/349. In this case, please consider including more information on the number of interviewers, their training and characteristics; and please provide the interview guide used.

Furthermore, please provide additional information regarding how informed consent was documented for the participants.

Reviewers' comments:

Reviewer's Responses to Questions

**Comments to the Author**

1. Is the manuscript technically sound, and do the data support the conclusions?

Reviewer #1: Yes

2. Has the statistical analysis been performed appropriately and rigorously? 

Reviewer #1: N/A

3. Have the authors made all data underlying the findings in their manuscript fully available?

Reviewer #1: Yes

4. Is the manuscript presented in an intelligible fashion and written in standard English?

Reviewer #1: No

5. Review Comments to the Author

Reviewer #1: Enablement of nursing home residents in infection prevention during GPs visits: A qualitative study

Thank you for the opportunity to review this interesting paper. I have made some general comments and specific comments about clarity and language. Hand hygiene as become an important area of interest with the COVID pandemic, and infection control in nursing homes even more important. I noticed that you haven’t made any reference to this in your discussions, or suggestions for further research. You may want to consider this to ensure relevancy.

Abstract

Line 19: ‘General practitioners avoid nosocomial infections’ makes it sound like the GP is avoiding getting an infection, rather than the residents. Consider changing to ‘prevent nosocomial infections’.

Line 31: ‘enablement’ instead of ‘enabling’

Line 33: ‘…. during the visit to the GP’ . Please clarify that the GP visited the resident rather than the other way round.

Line 34: Conclusion: the first sentence doesn’t make sense. Please consider rephrasing e.g. ‘lack of ability ..is due to’, or ‘For residents to maintain adequate hygiene… requires..’

Line 35: Please consider rephrasing the following sentence to ensure clarity: ‘The nursing home residents have a different need to be enabled..

Background

Line 49 ‘resistant’ rather than ‘resistance’

Line 50 Please write ‘staph’ in full

Line 52: bloodstream infections and pneumonia

Line 54: Besides – not quite the right word in this case. Please consider another word.

Line 56: specify EU member states

Line 56: remove to implement strategies (duplicated)

Line 55-68: Consider making into 2 sentences as rather long and poor punctuation.

Line 59 – remove comma after 2017

Line 63: Is hand rub better than washing hands with soap and water in this instance, or is it just the convenience of hand rub? Would be interesting to mention this here.

Line 72: GP (rather than GPs)

Line 73: Please consider clarifying sentence – ‘There is not GP with an overview…’

Line 79: Would be helpful to mention the guidelines you refer to in the discussion

Recruitment and informed consent:

121: Did only 12 or 250 GPs give consent to interview (if so, please comment on why you think this was the case, and the potential limitations). If more consented, please specify how they were sampled for interview.

Please give details about how residents gave their consent, to whom, and how it was recorded.

Data collection

Please consider adding what language the data was collected in.

Did the researchers have any relationship with the interviewees prior to the study?

130. Please specify here if the interviews were audio-recorded.

131: Please consider the following revision: prevention measures during GP visits, instead of while

131: Was an interview guide used? Please provide example questions. Did you pilot the interview guide before the interviews?

141: Please consider removing ‘and therapies’ or rewording the sentence.

Data analysis:

Please give details about how were the field notes from the resident interviews were used in the analysis

Please give details of how and when the data was translated to English.

149: Please specify who did the analysis (initials). Were they the same researchers who conducted the interviews?

151: It is not clear how the 5 major categories informed the themes described in the findings section. Please give more details about how you arrived at the final themes.

Please give details about consideration of data saturation.

Results

I would be helpful to orient the reader by listing the main themes first.

Please give some context to the quotations used (e.g. Female GP with *years experience)

172: Six main categories are now mentioned. Should these be themes? Also there are only 5.

176: Theme: this provides interesting context for the research but seems so move away from the point. I would consider shortening

179: ‘In case of questions’ – please give details about what sort of questions are being referred to. Questions from residents or questions from the GP?

201: This paragraph is talking about GPs views and behaviours to MRSA and should possibly be discussed within another theme.

221: quotation – it is unclear if this is a single conversation. Also the full conversation is not necessary to exemplify the findings. Consider shortening to make more succinct.

233: Please make clear what is meant by ‘rejecting microbiological differentiation’.

235: ‘without thinking about the effects of all residents’ – is this the authors interpretation of the GPs actions or was there more information than was given in the quotation?

237: Please consider truncating the quotation at line 240. The quotation after this point does not add anything further.

253: Please consider using another word than ‘overwhelmed’ which implies that the participants were not able to give an answer to the question.

271 and272 – please clarify why some words are in quotation marks. If they are quotations from interviews, please put in italics.

282: This quotation can also be truncated as the rest talks about how often the GP visits rather than the relationship

290: Consider rephrasing as follows: None of the residents described receiving hand hygiene training. Please consider adding details about why residents thought this to be the case.

308: Please consider using another term than ‘dared’ as this word suggests a high element of risk. Residents didn’t fell able to talk to their GP?

317: ‘GPs did not discuss infection prevention’ – please clarify sentence as GPs did discuss prevention in the quotation below.

Discussion

In general, I felt that the discussion section was not oriented around the main aims of the research: hand hygiene and patient enablement. You could consider structuring your discussion section around TPB. Further discussions on enablement would be helpful.

334: it is not clear how TPB was used in this study. Please give further details.

338 – 348: This paragraph discusses the context of GP visits to nursing homes but it not particularly relevant to the aims of the research. Please consider revising and linking these points to hand hygiene and whether you found any relationship with the types of visits and use of hand hygiene measures.

350-355: Please give some consideration about how the patient-practitioner relationship affects hand hygiene measures

355: Please comment about whether GPs were asked about their knowledge of hand hygiene regulations.

365: Discussion about previous antibiotics was not presented in the results section. Is this a new finding?

367: Please consider another term for ‘hardly perceived’

367: Please clarify who ‘They’ are (GPs, residents or both)

375: Remove ‘hand’.

Limitations

Did the authors consider interviewing the nursing home staff for their perspectives of hand hygiene measures in nursing homes?

382: whilst results couldn’t be generalised to residents with cognitive impairment, there is mention in the findings that a large majority of residents do have dementia. I think would be helpful to acknowledge this in the discussion too – and suggest what might be transferable, and what could be done in these cases.

Conclusions

394: should empowerment be enablement as they mean different things?

6. PLOS authors have the option to publish the peer review history of their article (what does this mean?). If published, this will include your full peer review and any attached files.

Reviewer #1: **Yes: **Jane Vennik

---

## [Author Response · Author response to Decision Letter 0]

11 Feb 2022

PONE-D-20-34748

'Response to Reviewers'

Revision: Enablement of nursing home residents in infection prevention during GPs visits: A qualitative study

16. September 2021

Dear Reviewer,

Thank you very much for taking the time and effort to read our manuscript. We appreciate your careful review and valuable suggestions. A revision of the paper has been carried out to take all of them into account. To better understand our modifications, we have highlighted them in yellow and provide an exact line number where we made the changes. Thank you very much for the helpful language corrections. We have implemented them all precisely in the manuscript. We have marked this with "implemented" in the following table. 

We have had the entire document professionally proofread.

Kind regards,

The Authors

---

## [Editor Report · Decision Letter 1]

23 Mar 2022

Enablement of Nursing Home Residents in Infection Prevention During General Practitioner Visits: A Qualitative Study

PONE-D-20-34748R1

Dear Dr. Ernstmann,

We’re pleased to inform you that your manuscript has been judged scientifically suitable for publication and will be formally accepted for publication once it meets all outstanding technical requirements.

Kind regards,

Monika Pogorzelska-Maziarz

Academic Editor

PLOS ONE
---

## [Editor Report · Acceptance letter]

29 Mar 2022

PONE-D-20-34748R1 

Enablement of Nursing Home Residents in Infection Prevention During General Practitioner Visits: A Qualitative Study 

Dear Dr. Ernstmann:

I'm pleased to inform you that your manuscript has been deemed suitable for publication in PLOS ONE. Congratulations! Your manuscript is now with our production department. 

Kind regards, 

on behalf of

Dr. Monika Pogorzelska-Maziarz 

Academic Editor

PLOS ONE